# Superior Biological Healing of Hamstring Autografts Compared with Tibialis Allografts after Anterior Cruciate Ligament Reconstruction: A Propensity Score Matching Analysis Based on Second-Look Arthroscopy

**DOI:** 10.3390/medicina60101631

**Published:** 2024-10-06

**Authors:** Seo-Jun Lee, Jun-Gu Park, Seung-Beom Han, Ji-Hoon Bae, Ki-Mo Jang

**Affiliations:** 1Department of Orthopaedic Surgery, Korea University Anam Hospital, Korea University College of Medicine, 73, Inchon-ro, Seongbuk-gu, Seoul 02841, Republic of Korea; lsjohn@korea.ac.kr (S.-J.L.); jgpark11@gmail.com (J.-G.P.); oshan@korea.ac.kr (S.-B.H.); 2Department of Orthopaedic Surgery, Korea University Guro Hospital, Korea University College of Medicine, 148, Gurodong-ro, Guro-gu, Seoul 08308, Republic of Korea; osman@korea.ac.kr

**Keywords:** anterior cruciate ligament, anterior cruciate ligament reconstruction, hamstring autograft, tibialis allograft, second-look arthroscopy

## Abstract

*Background and Objectives:* Remodeling and healing of the graft are crucial processes for long-term graft survival after anterior cruciate ligament reconstruction (ACLR). However, few studies have objectively evaluated the differences in graft healing between autografts and allografts. This study aimed to compare the status of the anterior cruciate ligament (ACL) grafts between hamstring tendon (HT) autografts and tibialis anterior tendon (TAT) allografts using second-look arthroscopy. *Materials and Methods:* The outcomes of 193 consecutive patients (153 males and 40 females, with an average age of 30.38 and BMI of 25.43 kg/m^2^) who underwent second-look arthroscopy following primary ACLR were retrospectively reviewed. Prior to participating in this study, all patients provided written informed consent. The patients were divided into two groups: those with HT autografts and those with TAT allografts. Confounding factors were matched between the two groups using propensity score matching (PSM). ACL graft status was assessed during second-look arthroscopy using a numeric scale system based on the degree of four parameters: graft tension, continuity, synovium coverage, and vascular marking. Clinical outcomes were assessed using the Lysholm and International Knee Documentation Committee (IKDC) scores. Graft status and clinical outcomes were compared between the two groups. Additionally, a subgroup analysis based on the timing of the second-look arthroscopy (12–24 months vs. >24 months after the initial ACLR) was conducted. *Results:* After PSM, 62 patients were included in each group. The second-look arthroscopy was conducted at 23.6 ± 6.6 months for the HT group and at 24.0 ± 7.9 months for the TAT group (*p* = 0.749). The continuity and tension of the ACL graft were not significantly different between the two groups (*p* = 0.146 and 0.075, respectively). However, the TAT group exhibited significantly inferior synovial coverage and vascular marking of the ACL graft compared with the HT group (*p* = 0.021 and 0.007, respectively). These findings were consistent regardless of the timing of the second-look arthroscopy. Clinical outcomes, according to the Lysholm and IKDC scores, significantly improved in both groups with no significant differences (*p* = 0.386 and 0.733, respectively). *Conclusions:* Although there were no differences in graft tension and continuity between HT autografts and TAT allografts, the biological healing of ACL grafts, in terms of synovialization and vascularization, was superior in HT autografts compared to TAT allografts.

## 1. Introduction

Anterior cruciate ligament (ACL) rupture is the most prevalent sports-related knee injury and often necessitates surgical intervention [1]. Extensive evidence indicates that a ruptured ACL does not heal spontaneously with nonoperative treatment [2]. Anterior cruciate ligament reconstruction (ACLR) is a widely accepted intervention that produces favorable results in physically active young individuals [3,4].

Various autograft and allograft tissues are used for primary ACLR [5]. The hamstring tendon (HT) autograft is a favored option because of its benefits such as minimal donor-site morbidity, rapid graft incorporation, and the absence of immune reactions and disease transmission risks. Nevertheless, in some patients, the small diameter of the tendons can reduce the tensile strength of the grafts [6,7,8,9]. Additionally, some studies have noted deficits in knee flexor strength and instances of saphenous nerve damage after ACLR with HT autografts [10,11]. Allografts are viable alternatives to autografts. The primary benefit of using allografts for ACLR is the ability to obtain grafts of the desired size without donor-site morbidities [12]. Additional benefits include reduced operative times and improved cosmetic outcomes [13]. Tibialis tendon allografts are increasingly preferred over Achilles tendon allografts because of their poor bone block quality [14]. Some controlled clinical studies have demonstrated comparable outcomes between soft-tissue allografts and HT autografts [15], while others have indicated that allograft tendons may exhibit inferior graft maturity, a higher risk of graft failure, and increased knee laxity compared to autograft tendons in ACLR [16,17].

There are two primary sites of healing in ACLR, each with distinct biological processes that must be independently evaluated: intra-articular graft remodeling, known as “ligamentization”, and intra-tunnel graft incorporation, referred to as “bone integration”, which occurs either through bone-to-bone or tendon-to-bone healing [18,19]. These healing processes are essential for long-term graft survival after ACLR. Irrespective of whether the tissue is an allograft or autograft, it undergoes a similar ligamentization process when used as an ACL substitute [20]. Previous studies have identified various methods for clinical and biological assessment of ACL graft healing at different stages. Clinically, imaging techniques, arthroscopy, and biopsy are used to observe transitions in tendon grafts post-ACLR. Concurrently, numerous histological and biochemical pre-clinical studies have revealed diverse molecular and cellular responses in tendon grafts following ACLR [21]. Previous studies reported that the quality of the graft ligamentization is strongly correlated with long-term graft survival [22,23].

Although second-look arthroscopy is invasive, it remains the most reliable method for thoroughly evaluating the ACL, including tension, synovial coverage, and graft continuity [24]. However, most studies employing second-look arthroscopy to assess the ACL graft status have focused on comparing objective outcomes based on surgical techniques and graft types.

Few studies have objectively evaluated differences in graft healing between autografts and allografts. Therefore, this study aimed to compare the status of ACL grafts between HT autografts and tibialis anterior tendon (TAT) allografts in terms of graft continuity, tension, synovial coverage, and vascularization using second-look arthroscopy after primary ACLR, employing propensity score matching (PSM) analysis. We hypothesized that following ACLR, the biological healing of ACL grafts would be superior in HT autografts to that in TAT allografts, based on PSM analysis.

## 2. Materials and Methods

### 2.1. Study Design and Patient Enrollment

After receiving approval from the Institutional Review Board (approval no. 2023AN0422), the medical records of 247 consecutive patients who underwent second-look arthroscopy after primary ACLR at our institute between January 2010 and December 2020 were retrospectively reviewed. Prior to participating in this study, all patients provided written informed consent. Patients were included if they underwent second-look arthroscopy at least one year after the primary ACLR. The exclusion criteria were as follows: (1) patients who underwent ACLR with a mixed graft (autologous tendon with augmented allograft); (2) those with bilateral ACL injuries; (3) those with concurrent injuries to other ligaments; (4) those who underwent revision ACLR; and (5) those with surgical site infection. After assessing their eligibility, 193 patients who underwent primary ACLR were enrolled in this study (Figure 1). The total number of cases was 193, consisting of 153 males and 40 females, with an average age of 30.38 years. The mean ages for the HT autograft and TAT allograft groups were 29.3 years and 31.7 years, respectively. As for BMI, the overall average was 25.43 kg/m^2^, with the mean BMI for the HT autograft and TAT allograft groups being 25.3 kg/m^2^ and 25.6 kg/m^2^, respectively.

### 2.2. Surgical Procedures

ACLR was performed by a single senior surgeon for patients presenting with ACL rupture and anterior and rotatory instabilities, as confirmed by magnetic resonance imaging (MRI) and physical examinations. After routine arthroscopic examination, a femoral tunnel was created at the anatomical footprint of the ACL by using an outside-in technique with a FlipCutter (Arthrex, Naples, FL, USA). Similarly, a tibial tunnel was created at the anatomical footprint of the ACL using a guide and reamer. The ACL graft was secured on the femoral side with TightRope^®®^ RT (Arthrex, Naples, FL, USA) and on the tibial side with a combination of an interference screw and a post-tie with a cancellous screw. In terms of tendon type, patients were thoroughly informed about the advantages and disadvantages of allografts and autografts prior to surgery and made their decision accordingly. Patients were divided into two groups: those with HT autografts and those with TAT allografts. For the HT autograft, the semitendinosus and gracilis tendons were harvested using a tendon stripper and a double-looped (four-stranded) graft was prepared. Double-stranded TAT allografts were prepared.

### 2.3. Clinical and Second-Look Arthroscopic Assessment

Subjective clinical outcomes were assessed using the Lysholm and International Knee Documentation Committee (IKDC) scores, both preoperatively and at the time of second-look arthroscopy.

Our research team performed second-look arthroscopy with patient consent when removing the cancellous screw after ACL surgery. During this procedure, we assessed the status of each graft to determine objective outcomes. During second-look arthroscopy, the ACL graft status was objectively evaluated using a numerical scale with the following four parameters: (1) ACL graft continuity, (2) graft tension, (3) vascular marking of the graft, and (4) synovial coverage [25,26,27,28,29,30]. Each subscale was rated on a scale of 0, 1, or 2 points, with scores assigned based on specific predefined criteria. Graft continuity was graded according to the extent of the tear as follows: no rupture, partial tear, or complete tear. Additionally, graft tension was evaluated based on the degree of elongation during probing and categorized as less than 3 mm, 3–5 mm, or >5 mm. Depending on the synovial coverage and vascular marking of the graft, it was graded as > 75%, 25–75%, or less than 25%. The overall status of the graft was determined by adding points from each subscale and categorizing them as Excellent (8–7 points), Good (6–5 points), Fair (4–3 points), or Poor (2–0 points) (Table 1). Our research team has been conducting evaluations on these four parameters during surgery since 2010, using a second-look approach following ACLR. Assessments were carried out through close observation and probing during second-look arthroscopy, and relevant findings were recorded immediately in the surgical records. From 2010 to the present, evaluations and recordings have been consistently performed by the same surgeon using the same method.

### 2.4. Statistical Analysis

In this study, it was not feasible to randomly assign the use of autografts and allografts to each patient. Instead, patients made their decisions after being thoroughly informed of the advantages and disadvantages of each graft. Despite this non-randomized situation, we aimed to elucidate the cause-and-effect relationship by using propensity scores to mimic the characteristics of a randomized controlled trial [31]. Matching analysis with propensity scores was performed to eliminate baseline differences in some factors during comparison. Propensity score matching (PSM) was used to minimize potential confounding factors and treatment selection bias and to adjust for significant differences in baseline covariates [32]. Confounding factors, including patient demographics, time to second-look arthroscopy, and combined meniscal tears, were matched between the two groups using PSM. Propensity score matching entails forming matched sets of treated and untreated subjects who share a similar value of the propensity score. In this process, propensity scores were used to perform 1:1 matching between 106 HT and 87 TAT subjects with similar propensity scores [31]. After PSM, 62 patients were selected from each group (Figure 1). Graft status and clinical outcomes were compared between the two groups using an independent *t*-test for continuous variables. Additionally, a subgroup analysis based on the timing of the second-look arthroscopy (12–24 months vs. >24 months after the initial ACLR) was conducted to compare the outcomes between these groups. Statistical analyses were performed using the IBM SPSS Statistics 21.0 software (SPSS, Armonk, NY, USA). Statistical significance was set at *p* < 0.05.

## 3. Results

### 3.1. Patient Demographics

A total of 193 patients were enrolled and analyzed in our study. Of these, 153 were male and 40 were female, with an overall mean age of 30.38 years and an average BMI of 25.43 kg/m^2^. Among the 193 patients, 106 opted for a hamstring tendon (HT) autograft, while 87 selected a tibialis anterior tendon (TAT) allograft. The average time to second-look arthroscopy was 25.03 months, and 105 cases involved concomitant meniscus injuries (Table 2).

### 3.2. PSM

After the PSM, 62 patients were included in each group. Before PSM, the time to second-look arthroscopy after ACLR differed significantly between the two groups (*p* = 0.004). After matching, the demographic variables (i.e., age and sex), body mass index (BMI), time to second-look arthroscopy, cases with combined meniscal tears, Lysholm score, and IKDC subjective knee function score showed no statistical differences. The second-look arthroscopy was conducted at 23.6 ± 6.6 months (12–49 months) for the HT autograft group and at 24.0 ± 7.9 months (14–57 months) for the TAT allograft group, with no significant difference between the groups (*p* = 0.749) (Table 3).

### 3.3. Comparison of Status of ACL Grafts between HT Autograft and TA Allograft

The total score for the four parameters regarding ACL status was 7.5 ± 1.0 for the HT group and 6.6 ± 1.9 for the TAT group. The total score of the TAT group was significantly lower than that of the HT group (*p* = 0.001). Considering each parameter, the ACL graft continuity score was 1.9 ± 0.3 for the HT group and 1.8 ± 0.4 for the TAT group. In terms of ACL graft tension, the score was 2.0 ± 0.2 for the HT group and 1.9 ± 0.4 for the TAT group. The continuity and tension of the ACL graft were not significantly different between the two groups (*p* = 0.146 and *p* = 0.075, respectively). However, the TAT group exhibited significantly inferior synovium coverage (1.4 ± 0.7 for TAT and 1.8 ± 0.5 for HT) and vascular marking (1.4 ± 0.7 for TAT and 1.8 ± 0.4 for HT) of the ACL graft compared to the HT group (*p* = 0.005 and 0.001, respectively). (Table 4). Regarding the distribution of the two different types of grafts, the TAT group exhibited significantly inferior synovial coverage and vascular marking of the ACL graft compared to the HT group (*p* = 0.021 and 0.007, respectively) (Figure 2).

### 3.4. Subgroup Analysis between HT Autograft and TA Allograft during the Early Period

Subgroup analysis was performed for each graft group between the early period (time to second-look arthroscopy in 12–24 months after primary ACLR) and late period (time to second-look arthroscopy >24 months after primary ACLR). In the HT autograft group (N = 62), there were 44 and 18 patients in the early and late periods, respectively. In the TAT allograft group (N = 62), there were 40 patients in the early period and 22 in the late period. Regardless of the timing of second-look arthroscopy, the TAT group exhibited significantly inferior synovial coverage (*p* = 0.048 for the early period and *p* = 0.045 for the late period) and vascular marking (*p* = 0.025 for the early period and *p* = 0.030 for the late period) of the ACL graft compared to the HT group (Table 5 and Table 6).

We compared the clinical outcomes between the two groups using Lysholm and IKDC scores. Clinical scores were measured twice to assess changes over time, preoperatively and at the time of second-look arthroscopy. This measurement was performed across all groups. As expected, the clinical scores improved over time in both groups; however, no statistically significant differences were observed between the two groups (*p* = 0.386 and 0.733, respectively) (Table 7).

## 4. Discussion

The most important finding of the current study was that the biological healing of ACL grafts, in terms of synovialization and vascularization, was superior in HT autografts compared to TAT allografts, although there were no differences in graft tension and continuity between the HT autografts and TAT allografts. These findings were consistent regardless of the timing of the second-look arthroscopy. There were no significant differences in Lysholm and IKDC scores between the groups.

Several studies have evaluated the status of ACL grafts by using second-look arthroscopy. Bottoni et al. found that knees reconstructed with a hamstring autograft demonstrated statistically significant graft survivability compared to knees reconstructed with a tibialis posterior allograft [22]. In a study by Yoo et al. that compared autografts and allografts using second-look arthroscopy following ACLR, it was reported that there was no significant difference in graft continuity between the two types of grafts. However, there was a significant difference in synovial coverage, indicating biological healing [30]. In our study, the continuity and tension of the ACL graft were not significantly different between the two groups. However, the allograft group exhibited significantly inferior synovial coverage (1.4 ± 0.7 for the TAT allografts vs. 1.8 ± 0.5 for the HT autografts) and vascular marking (1.4 ± 0.7 for the TAT allografts vs. 1.8 ± 0.4 for the HT autografts) of the ACL graft compared to the autograft group. In this study, the PMS was conducted to eliminate baseline differences.

To understand the significant differences between allografts and autografts in the four parameters indicating graft status, it is essential to examine the graft healing process. Graft healing involves two distinct processes, bone integration and biological healing. Bone integration comprises the initial inflammatory phase, the osteointegration (proliferation phase) phase, and maturation (remodeling phase) phase. The inflammatory phase occurs immediately after surgery and lasts for several days. This phase involves inflammation at the bone–tendon interface, which helps clean the surgical area and initiate the healing process. Second, the proliferation phase occurs approximately 1–6 weeks after surgery. During this phase, osteoblasts lay down a new bone matrix and progressively integrate the graft into the bone tunnel. Third, the remodeling phase spans 2–6 months postoperatively, during which the graft becomes firmly anchored, gaining strength and stability. Biological healing begins shortly after surgery and can extend for over a year, progressing through necrosis, revascularization, cellular proliferation, and remodeling. However, bone integration generally occurs first as it is a crucial step for the initial stability and anchoring of the graft, setting the stage for subsequent biological healing processes [18,33,34].

Second-look arthroscopy was performed to observe four aspects of the graft (graft tension, graft continuity, synovial coverage, and vascular marking) in the present study. Graft tension and continuity are closely associated with bone integration, whereas synovial coverage and vascular markings are closely related to biological healing. Therefore, starting second-look arthroscopy one year after surgery likely influenced the results more in terms of biological healing than bone integration. In this study, autografts were found to be superior to allografts in terms of their biological aspects. Based on the above these results, we could consider further research that could include the analyzing of hybrid grafts that were excluded based on the exclusion criteria, in addition to comparing autografts and allografts.

Several factors contributed to this superiority. The first pertains to the immune response. Autografts, sourced from the patient’s own body, provoke a minimal immune reaction, facilitating superior integration and healing. Conversely, donor-derived allografts are more prone to eliciting an immune response, potentially hindering healing and leading to graft rejection or delayed incorporation. Second, differences in revascularization and cellular repopulation play a role. Autografts typically undergo more rapid and effective revascularization than allografts, benefiting from immediate revascularization and repopulation by host cells. This accelerated process enhances the nutrient and oxygen supply, promoting quicker and more robust healing. By contrast, allografts often require longer periods of revascularization, potentially delaying healing. Third, the risk of disease transmission differs. Autografts carry a negligible risk of disease transmission as they originate from the patient’s own tissue. However, despite stringent screening and processing, allografts may carry a low risk of transmission of infectious diseases [4,35].

Some studies that have performed second-look arthroscopy after ACL reconstruction surgery and reported that knee functional assessments (such as Lysholm, IKDC, and Tegner activity scores) conducted regardless of the type of graft used during surgery showed no statistically significant differences in clinical outcomes among patients [5,30,36]. The present study demonstrated findings similar to those of previous studies. Additionally, studies have indicated that psychological factors and rehabilitation programs, rather than the type of graft used, may influence clinical outcomes. Ardern et al. suggested that psychological responses before surgery and early recovery are associated with a return to sports [37]. Buckthrope et al. reported that recovering power of the hamstring muscles is an important process after ACL reconstruction [38]. Muller et al. suggested that the single-hop distance and ACL-RSI were the strongest predictive parameters for return to sports following ACLR. Jeon et al. found that younger age and thigh muscle strength were factors that determined substantial clinical benefits after ACLR [39].

Our study has some limitations. First, the current study was conducted retrospectively and included only patients who underwent second-look arthroscopy. Therefore, there may have been an inherent selection bias. Second, the second-look arthroscopic evaluations were performed by the same surgeon who performed the initial ACLR. Due to the limitation of conducting multiple procedures on a single patient, obtaining inter-observer and intra-observer reliability was inherently unfeasible, which may lead to assessment biases. Third, during PSM, there may have been factors influencing graft healing that were not accounted for, potentially introducing bias by omitting variables that did not affect graft healing. Fourth, we observed the ACL graft status at different time points during second-look arthroscopy (12–24 months vs. >24 months). However, this comparison may have limited significance, as the PSM already included the variable of time to second-look arthroscopy, thereby potentially diminishing the meaningfulness of comparing the early and late periods based on the graft type. Finally, the number of enrolled patients was relatively small.

## 5. Conclusions

Although there were no differences in graft tension and continuity between the HT autograft and TAT allograft groups, the biological healing of ACL grafts in terms of synovialization and vascularization was superior in the HT autograft group compared to the TAT allograft group.

## Figures and Tables

**Figure 1 medicina-60-01631-f001:**
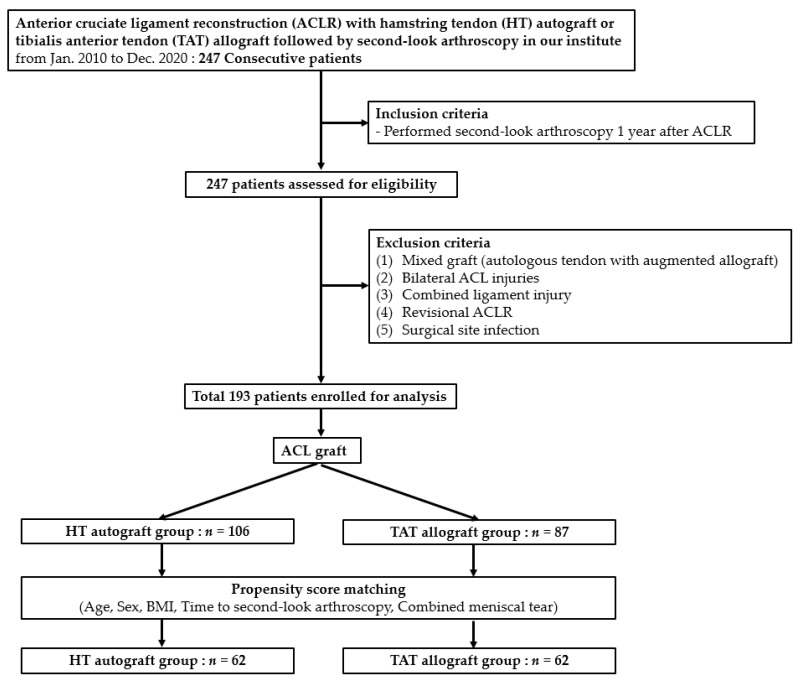
Flowchart of patient selection in the present study.

**Figure 2 medicina-60-01631-f002:**
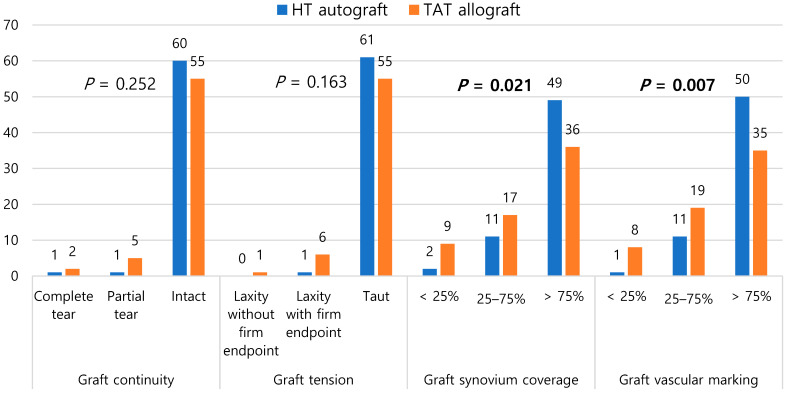
Distribution of HT autograft and TAT allograft based on graft continuity, tension, synovial coverage, and vascular marking during second-look arthroscopy following ACLR. HT, hamstring tendon; TAT, tibialis anterior tendon.

**Table 1 medicina-60-01631-t001:** ACL graft evaluation scale during second-look arthroscopy.

Score	ACL Graft Status on Second Look Arthroscopy			
	Graft Continuity	Graft Synovium Coverage	Graft Tension	Graft Vascular Marking
2	No tear	>75%	Elongation 0–2 mm on probing (Taut)	>75%
1	Partial tear	25–75%	Elongation 3–5 mm on probing (Laxity with firm end point)	25–75%
0	Complete tear	<25%	Elongation >5 mm on probing (Laxity without firm end point)	<25%

Total maximum score is 8. ACL, anterior cruciate ligament.

**Table 2 medicina-60-01631-t002:** Characteristics of the patients enrolled for analysis.

Total Patients	
Number	193
Age (yr)	30.38
Sex (M/F)	153/40
BMI (kg/m^2^)	25.43
Graft type	
HT autograft	106
TAT allograft	87
Time to second-look arthroscopy (m)	25.03
Combined meniscus tear (n)	105

HT, hamstring tendon; TAT, tibialis anterior tendon; BMI: body mass index.

**Table 3 medicina-60-01631-t003:** Comparison of patient characteristics between two groups with HT autograft and TAT allograft before and after propensity score matching ^a^.

	Total Cases (*n* = 193)			Propensity Score Matched Cases (*n* = 124)		
Variables	HT Autograft	TAT Allograft	*p*-Value	HT Autograft	TAT Allograft	*p*-Value
N	106	87		62	62	
Age, y	29.3 ± 10.8 (15–56)	31.7 ± 11.4 (16–66)	0.127	29.9 ± 11.4 (15–56)	31.3 ± 11.4 (17–66)	0.485
Sex, male/female	88/18	65/22	0.211	50/12	47/15	0.514
BMI, kg/m^2^	25.3 ± 3.5 (18.1–34.6)	25.6 ± 4.2 (17.5–40.0)	0.563	25.2 ± 3.5 (18.1–34.6)	25.1 ± 3.6 (18.8–33.5)	0.879
Time to second-look arthroscopy, m 12–24 months >24 months	26.7 ± 10.3 (12–85) 57 49	23.0 ± 7.3 (12–57) 62 25	**0.004**	23.6 ± 6.6 (12–49) 44 18	24.0 ± 7.9 (14–57) 40 22	0.749
Combined meniscus tear, n	62/106	43/87	0.246	28/62	33/62	0.369

^a^ Values are presented as n (number) or mean ± SD (range). HT, hamstring tendon; TAT, tibialis anterior tendon; BMI: body mass index.

**Table 4 medicina-60-01631-t004:** Comparison of status of ACL grafts during second-look arthroscopy between HT autograft and TAT allograft.

	HT Autograft (*n* = 62)	TAT Allograft (*n* = 62)	*p*-Value
Second look findings			
ACL graft continuity	1.9 ± 0.3	1.8 ± 0.4	0.146
ACL graft tension	2.0 ± 0.2	1.9 ± 0.4	0.075
ACL graft synovial coverage	1.8 ± 0.5	1.4 ± 0.7	**0.005**
ACL graft vascular marking	1.8 ± 0.4	1.4 ± 0.7	**0.001**
Total score	7.5 ± 1.0	6.6 ± 1.9	0.001
Overall status			0.009
Excellent (8–7)	54	39	
Good (6–5)	6	12	
Fair (4–3)	2	9	
Poor (<3)	0	2	

ACL, Anterior cruciate ligament; HT, Hamstring tendon; TAT, Tibialis anterior tendon.

**Table 5 medicina-60-01631-t005:** Subgroup analysis comparing the status of ACL graft on second-look arthroscopy between HT autograft and TAT allograft during the early period.

	Early Period (Time to Second-Look Arthroscopy, 12–24 Months)		
	HT Autograft (*n* = 44)	TAT Allograft (*n* = 40)	*p*-Value
Second look findings			
ACL graft continuity	1.9 ± 0.3	1.9 ± 0.4	0.684
ACL graft tension	2.0 ± 0.2	1.9 ± 0.4	0.234
ACL graft synovial coverage	1.8 ± 0.5	1.5 ± 0.8	**0.048**
ACL graft vascular marking	1.8 ± 0.4	1.6 ± 0.7	**0.025**
Total score	7.5 ± 1.0	6.9 ± 1.9	0.047
Overall status			0.116
Excellent (8–7)	39	29	
Good (6–5)	4	5	
Fair (4–3)	1	5	
Poor (<3)	0	1	

ACL, anterior cruciate ligament; HT, hamstring tendon; TAT, tibialis anterior tendon.

**Table 6 medicina-60-01631-t006:** Subgroup analysis comparing the status of ACL graft on second-look arthroscopy between HT autograft and TAT allograft during the late period.

	Late Period (Time to Second-Look Arthroscopy, >24 Months)		
	HT Autograft (*n* = 18)	TAT Allograft (*n* = 22)	*p*-Value
Second look findings			
ACL graft continuity	2.0 ± 0.0	1.8 ± 0.5	0.060
ACL graft tension	2.0 ± 0.0	1.8 ± 0.4	0.060
ACL graft synovial coverage	1.7 ± 0.6	1.3 ± 0.7	**0.045**
ACL graft vascular marking	1.7 ± 0.6	1.2 ± 0.7	**0.030**
Total score	7.4 ± 1.1	6.1 ± 1.8	0.008
Overall status			0.012
Excellent (8–7)	15	6	
Good (6–5)	2	7	
Fair (4–3)	1	4	
Poor (<3)	0	1	

ACL, anterior cruciate ligament; HT, hamstring tendon; TAT, tibialis anterior tendon.

**Table 7 medicina-60-01631-t007:** Comparison of clinical outcomes between HT autograft and TAT allograft.

	HT Autograft (*n* = 62)	TAT Allograft (*n* = 62)	*p*-Value
Lysholm score			
Preoperative	57.8 ± 17.3	53.4 ± 21.4	0.371
At the time of second-look arthroscopy	86.6 ± 11.7	84.6 ± 12.9	0.386
IKDC subjective knee function score			
Preoperative	47.20 ± 15.3	49.5 ± 16.3	0.575
At the time of second-look arthroscopy	79.7 ± 16.1	80.8 ± 15.0	0.733

HT, hamstring tendon; TAT, tibialis anterior tendon; IKDC, International Knee Documentation Committee.

## Data Availability

The data presented in this study are available on request from the corresponding author.

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
