# Peer review of "Superior Biological Healing of Hamstring Autografts Compared with Tibialis Allografts after Anterior Cruciate Ligament Reconstruction: A Propensity Score Matching Analysis Based on Second-Look Arthroscopy"

_medicina, 2024, doi:10.3390/medicina60101631_

Round 1

Reviewer 1 Report (New Reviewer)

Comments and Suggestions for Authors

This study aimed to compare the status of ACL grafts between HT autografts and tibialis anterior tendon (TAT) allografts in terms of graft continuity, tension, synovial coverage, and vascularization using second-look arthroscopy. We hypothesized that the biological healing of ACL grafts would be superior in HT autografts than in compared to TAT allografts.

The manuscript is well structured, deals with a topic of current interest and is of potential interest to the scientific community. I have only a few suggestions for the authors.

The abstract is too long and should be re-modulated to provide only the relevant information.

In particular, regarding the introduction, the authors should better specify the purpose of the study in the final part. They should also provide their research hypothesis.

The authors should include a summary table of the characteristics of the subjects recruited for the study.

Even with regards to the results, it would be appropriate to insert some graphs or tables in order to make this section more readable.

The formatting of bibliographic references in the text should be reviewed. References should be inserted before the full stop.

Author Response

Reviewer 2 Report (Previous Reviewer 1)

Comments and Suggestions for Authors

The resubmitted manuscript has been constructed on the comparison of autograft (hamstring tendon (HT)) and allograft (tibialis anterior tendon (TAT)) by assessing the Lysholm and International Knee Documentation Committee  (IKDC) scores obtained during second-look arthroscopy of anterior cruciate ligament reconstruction (ACLR). The evaluation parameters for the comparison of both grafts were graft continuity, tension, synovial coverage, and vascularization.

The article has a research plan which had been properly constructed and conducted, and the results associated with this research plan were presented properly. Regarding the novelty of the subject (as previously mentioned), authors represent the well-known characteristics of autografts and allografts by post re-surgical clinical evaluations, and therefore the novelty is limited. But still the results and discussions raised by the article worth to read. The introduction part covers the related information with appropriate references cited. Discussions were made by citing related information reported in the literature, and those seem to be appropriate.

I think these two points still needs to be clear.

- As authors defined, four parameters were investigated in this article, but it is not clear that how these parameters had been measured for each patient. Were they only observed by naked eye or only one person scored all patients’ data? If so, how reliable are these observations laying between 2010 to 2020?

- There is no indication of the short- and long-term medication prescribed for patients (immunosuppressive, etc.) and its potential effect investigated parameters.

Comments on the Quality of English Language

The language is fine and minor editing can be applied.

Author Response

Reviewer 3 Report (Previous Reviewer 2)

Comments and Suggestions for Authors

In this study, In this study, the comparation between hamstring autografts and tibialis allografts after anterior cruciate ligament reconstruction has been investigated. The previous comments have been modified to some extent, Although various factors such as immune system and genetic factors as well as surgical conditions and..., are effective. , but they are acceptable.

Author Response

Comments 1 : In this study, the comparation between hamstring autografts and tibialis allografts after anterior cruciate ligament reconstruction has been investigated. The previous comments have been modified to some extent, Although various factors such as immune system and genetic factors as well as surgical conditions and..., are effective. , but they are acceptable.

Author response) Thank you for your thoughtful review and comment. In future studies comparing hamstring autografts and tibialis anterior allografts after ACL reconstruction, we will aim to incorporate the consideration of patients' immune systems and genetic factors.

This manuscript is a resubmission of an earlier submission. The following is a list of the peer review reports and author responses from that submission.

Round 1

Reviewer 1 Report

Comments and Suggestions for Authors

The manuscript is based on the comparison of autograft (hamstring tendon (HT)) and allograft (tibialis anterior tendon (TAT)) by assessing the Lysholm and International Knee Documentation Committee  (IKDC) scores obtained during second-look arthroscopy of anterior cruciate ligament reconstruction (ACLR). The parameters of evaluation are graft continuity, tension, synovial coverage, and vascularization.

The article submitted has a research plan which had been properly constructed and conducted, and the results associated with the research plan were presented. Regarding the novelty of the subject, authors represent the well known characteristics of autografts and allografts by post re-surgical clinical evaluations, and therefore the novelty is limited. But still the results and discussions raised by the article worth to read. The introduction part covers the related information with appropriate references cited. Discussions and conclusion are supported by data obtained and citations.  

Some points to be addressed by the authors as follows;

- Four parameters were investigated in this article but it is not clear that how these parameters had been measured for each patient. Were they only observed by naked eye? If so, how reliable are these observations laying between 2010 to 2020?

- There is no indication of the short- and long-term medication prescribed for patients (immunosuppressive, etc.)

- The abbreviation of the allograft shall be fixed as TAT or TA, they differ in the text, tables and figures.

- Statistical methods are not described in the text

Comments on the Quality of English Language

The language is satisfactory but needs a check for sentence structures.

Reviewer 2 Report

Comments and Suggestions for Authors

-          In this study, the comparative between hamstring autografts and tibialis allografts after anterior cruciate ligament reconstruction has been investigated. Although various factors such as immune system and genetic factors, as well as surgical conditions and..., are effective.

-            However, good information has been reported in this study, and it is an interesting study, but some points are suggested:

-          1. In this study only patients who underwent second-look arthroscopy were included in the study and this is one of the weaknesses of the study, which weakens the final result. Nevertheless, it is a good study.

-          It would have been better to investigate and report complications such as tunnel widening etc

Reviewer 3 Report

Comments and Suggestions for Authors

This study uses second look arthroscopy to compare an autograft with an allograft. Very interesting in deed! Allografts are gaining popularity especially in the United States , however there are clear disantavantages in their use, some of them are noted by the authors in the manuscript.

However there are quite a few methodological flaws detected.

First of all - is second look arthroscopy a routine in this institute or was it performed for the purposes of this study? This has to be made clear. This should be a prospective study with the second look performed at the same time (mark) in all included pts.

Then several ethical concerns are raised. Were the pts randomized? How was the decision for which graft to use, made? For example,  why use an allograft in a 15 year old patient? This needs to be explained.

Written informed consent taken from all pts should be noted in the abstract and M&Ms

Then the Propensity Score matching tool. Was this validated in any way? This need to be explained as well as how and why from 106 HT and 87 TAT the included pts were 62 in each group. Table 2 is very busy and confusing.

Age and time of second look scope disparity are huge. Regardles the subgroup analysis this creates a huge bias. The authors state that in their limitations

Graft survivorship should be presented in a proper way such as a Caplan Meyer  curve. 

Age,sex, BMI sould be mentioned in the abstract and M&Ms. 

English language review is required.

Comments on the Quality of English Language

Spelling and grammatical erros. More fluent English writting style is required.